# A 6-month, prospective randomized controlled trial of the TargetEd MAnageMent (TEAM) intervention vs. enhanced treatment as usual among Ugandans at risk for stroke

Mark Kaddumukasa[1], Martin Kaddumukasa[2], Scovia Nalugo Mbalinda[3], Josephine Najjuma[4], Jane Nakibuuka[2], Doreen Birungi[1], Carla Conroy[5,6], Joy Yala[5,6], Levicatus Mugenyi[1], Christopher J. Burant[7,8], Shirley Moore[8], Elly T. Katabira[1], Martha Sajatovic[6,9]*

1 Department of Medicine, School of Medicine, College of Health Sciences, Makerere University, Kampala, Uganda, 2 Department of Medicine, Mulago Hospital, Kampala, Uganda, 3 Department of Nursing, College of Health Sciences, Makerere University, Kampala, Uganda, 4 Department of Nursing, Mbarara University of Science and Technology, Mbarara, Uganda, 5 University Hospitals Cleveland Medical Center, Cleveland, Ohio, United States of America, 6 School of Medicine, Case Western Reserve University, Cleveland, Ohio, United States of America, 7 Frances Payne Bolton School of Nursing, Case Western Reserve University, Cleveland, Ohio, United States of America, 8 Veterans Affairs Northeast Ohio Healthcare System, Geriatric Research Education, and Clinical Center, Cleveland, Ohio, United States of America, 9 Neurological and Behavioral Outcomes Center, University Hospitals Cleveland Medical Center, Cleveland, Ohio, United States of America

* martha.sajatovic@uhhospitals.org

## Abstract

### Background

Among low and middle-income countries, especially in Sub-Saharan Africa, the stroke burden is severe, with increasing trends in stroke incidence, prevalence, and mortality.

### Aims

This 6-month, prospective randomized controlled trial (RCT) compared a novel stroke risk reduction approach (TargetEd manAgeMent Intervention (TEAM)) vs. Enhanced Treatment as Usual (ETAU) in 247 Ugandans at risk for stroke.

### Methods

Participants, enrolled across 3 Ugandan sites, were adults with high stroke risk. The primary outcome was a change in systolic blood pressure (SBP) from baseline to 6-month follow-up. Secondary outcomes included changes from baseline to 6 months on diastolic BP (DBP), serum lipids, glycosylated hemoglobin (HbA1c), self-efficacy and stress.

**Data availability statement:** All relevant data are within the paper and its Supporting Information files.

**Funding:** Martha Sajatovic and Elly Katabira received the award. "Research reported in this publication was supported by the National Institute Of Neurological Disorders And Stroke of the National Institutes of Health under Award Number R01NS118544. The content is solely the responsibility of the authors and does not necessarily represent the official views of the National Institutes of Health.

**Competing interests:** Dr. Sajatovic has research grants from Neurelis, Intra-Cellular, Merck, Otsuka, Teva and Alkermes, is a consultant to Alkermes, Otsuka, Janssen, Lundbeck, Teva and Neurelis and has received publication royalties from Springer Press, Johns Hopkins University Press, Oxford Press, and UpToDate. The other authors have nothing to report.

**Abbreviations:** BP, blood pressure; ETAU, enhanced treatment as usual; LMICs, low-middle income countries; NCDs, non communicable diseases; PEDs, peer educator dyads; RCT, randomized controlled trial; SAB, stakeholder advisory board; TEAM, Targeted MAnegeMent Intervention program.

## Results

The mean sample age was 55.4 (±SD = 12.0) with majority being women (n = 168, 68%). In addition to hypertension, the most common risk factors were hyperlipidemia (n = 199, 80.6%) and obesity (n = 98, 39.7%). Overall mean SBP and DBP at baseline were 143.0 (SD = 19.8, range 94.5–206) and 89.3 (SD = 14.0, range 61–136) respectively. In TEAM, SBP significantly improved from 145.7 (±21.5) at baseline to137.4 (±18.1) at 6-months vs. change from 141.9 (±18.4) to 141.1 (±21.9) for ETAU (p = 0.031). There were similar reductions in DBP favoring TEAM (p = .012). Compared to ETAU, TEAM showed improved physical activity (p = .017), self-efficacy (p < .001) and stress (p = .014).

## Conclusions

There is a need for effective and practical approaches to reduce stroke burden in Sub-Saharan Africa. Inclusion of the TEAM approach in primary care seems to be a pragmatic and effective way to potentially reduce stroke burden in Uganda.

## Trial registration

ClinicalTrials.gov identifier: NCT04685408, registered on 28 December 2020.

## Introduction

Stroke is a widely prevalent condition that imposes substantial burden on individuals and families, including disability and premature mortality. The 2019 Global Burden of Disease Study estimated that more than 12 million new strokes and over 6 million deaths occurred globally [1]. Stroke related disability-adjusted life years (DALYs) are also high with an estimated 143 million years of healthy life lost each year [1]. Among low and middle-income countries (LMICs), especially in Sub-Saharan Africa (SSA), stroke burden is particularly severe with increasing trends in stroke incidence, prevalence, and mortality [2]. Stroke crude incidence rates have risen from an average of 53 (range: 26–101) cases/100,000 population between 1973 and 1991–88 (range: 25–149) cases/100,000 population between 2003 and 2011 [2].

To reduce stroke burden in SSA, both primary and secondary prevention are important. Primary stroke prevention hinges on at-risk individuals possessing the awareness and knowledge to seek timely and preventative care. However, basic stroke knowledge may be sub-optimal as demonstrated in a population-based Ugandan survey that found nearly 75% of respondents unaware of any warning signs of stroke [3]. In SSA, stroke occurs at younger ages compared to Western countries [4]. As is the case globally, hypertension, diabetes, dyslipidemia, and lifestyle factors such as smoking, obesity, obstructive sleep apnea, physical inactivity, excessive alcohol consumption, and unhealthy diet are major modifiable risk factors [4,5]. The overall prevalence of hypertension in Uganda is relatively high at 26.4% of adults [6] and represents an actionable target to reduce stroke burden.

Secondary stroke prevention in SSA is also challenging given limited availability of rehabilitation and community support systems for stroke survivors and caregivers are often not available. Researchers in Ghana reported mild to moderate stigma in 80% of stroke survivors, with the degree of stigma directly related to the patient's degree of dependency [7]. Taken together, community-based primary and secondary stroke prevention strategies are urgently needed in LMICs including countries in SSA [8].

The **T**arget**E**d man**A**ge**M**ent (TEAM) Intervention is a curriculum-based self-management approach which uses nurse and peer educator dyads (PEDs) composed of patients and their care partners to co-deliver the intervention. A key principle of the TEAM program is on self-empowerment and being able to communicate with care providers to help manage stroke risk. A 6-month, uncontrolled, prospective pilot study to establish feasibility, acceptability and preliminary efficacy of TEAM in Ugandans at high risk for stroke showed TEAM participants had significant reductions from baseline in mean systolic blood pressure (SBP) and in total serum cholesterol to 24-week's levels [9].

As a follow-up to promising pilot findings, the TEAM approach was recently tested in a larger-scale randomized controlled trial (RCT) to evaluate primary effects on SBP and other key outcomes relevant to stroke risk including diastolic BP (DBP), cholesterol/lipids, glycosylated hemoglobin (HbA1c), body mass index (BMI) as well as measures that evaluated diet, activity levels, self-efficacy and stress. We hypothesized that TEAM would lead to improved outcomes on BP and other biological and behavioral stroke risk factors, specifically, 1) Individuals in TEAM would have significantly reduced SBP vs. individuals in enhanced treatment as usual (ETAU), 2) Individuals in TEAM would have greater reductions in total serum cholesterol compared to ETAU and 3) Individuals with diabetes in TEAM will have improved glycemic control as measured by serum HbA1c compared to ETAU.

## Methods

### Overview

This was a prospective RCT of 2 stroke reduction approaches (**T**arget**E**d man**A**ge**M**ent Intervention (TEAM) vs. ETAU in 247 Ugandans at risk for stroke. The start and end dates of recruitment and 6 months follow up were 06/01/2022 and 20/06/2024 respectively. Details of the study protocol, including mixed-methods work to refine the TEAM intervention prior to RCT implementation, have been published [10]. The project was conducted across 3 Ugandan sites, Makerere University College of Health Sciences (MakCHS) Mulago Hospital, Mbarara University of Science and Technology (MUST) Mbarara Hospital and Uganda Martyrs University (UMU) Nsambya Hospital and enrolled a representative sample of high-risk Ugandans. After screening and study baseline procedures, individuals were randomly assigned to receive either TEAM or ETAU. The primary RCT outcome was change in SBP from baseline to 6-months. Additional outcomes were changes from baseline to 6 months on DBP, serum cholesterol/lipids and HbA1c, BMI, diet and activity levels, self-efficacy, stress and medication adherence. Each participant was assessed 4 times: at screening, at baseline, at 13 weeks, and at 6 months.

### Sample recruitment

All individuals were ≥ 18 years of age and deemed at risk for stroke based upon the following: a.) High SBP defined as ≥140 mmHg assessed on at least 2 occasions at least 3 days apart and either b.) 1 other modifiable stroke risk factor including: diabetes, hyperlipidemia, obesity, smoking, problem alcohol use or sedentary lifestyle or c.) History of stroke or transient ischemic attack (TIA) within the past 5 years. The screening process for stroke risk factors [11] was based upon an initial self-report of status with respect to diabetes, hyperlipidemia, obesity, smoking, problem alcohol use or sedentary lifestyle as well as initial BP measurement. Individuals with 2 or more confirmed stroke risk factors then completed baseline evaluation study. The sample was selected to be broadly generalizable. Only individuals with dementia, sickle-cell disease, those who were pregnant or unable to participate in study procedures were excluded. All participants provided written informed consent.

## Ethics statement and ethical approval

The study was approved by the following two IRB committees from Case Western Reserve University (CWRU) IRB-STUDY20200882 and Makerere University, School of Medicine, Research and Ethics Committee (SOMREC) Mak-SOMREC-2020–179. Regulatory approval was from the Uganda National Council of Science and Technology; UNCST HS2944ES and registered on ClinicalTrials.gov identifier: NCT04685408 on December 28, 2020. Written informed consent was obtained from the study participants before enrolment into the study.

## Randomization

Computer-generated randomization allocated (1:1 basis) assignment to either TEAM or ETAU. Block randomization sizes ranging randomly between 4–8 helped to ensure that equal numbers of TEAM and ETAU patients occurred within strata and were balanced with respect to diabetes and previous stroke.

## Interventions

TEAM: Informed by social cognitive theory [12], TEAM uses nurses and peer educator dyads (PEDs) composed of patients who had suffered a stroke or transient ischemic attack and their immediate care partners to provide support to co-deliver the intervention. The content focused on critical post stroke care for the patient and care partner, problem solving and emotional management. These peer educator dyads received training with the nurse educators on how to conduct the intervention sessions.

TEAM began with one 60-minute 1:1 orientation session. This is followed by 6 hour-long group sessions with 6–8 patients and their care partners. The first orientation session was completed approximately 1 week post baseline, followed by group sessions done at 2, 4, 6, 8, 10 and at 12 weeks post-baseline. Individuals who missed a group session were encouraged to complete a make-up session. To reinforce learning after the TEAM group sessions were done, 3 brief (approximately 10–20 minute) monthly telephone calls occurred between the nurse and the patient over the next 3 months. All TEAM participants continued in treatment with their regular medical care providers and TEAM visits took place in these clinics. For this RCT the team hired and trained 2 nurses and 4 PEDs

ETAU: ETAU consisted of an orientation visit (approx. 30 minutes) with a nurse who provided verbal and written education materials on stroke risk such as hypertension, obesity, high salt/high fat diet and diabetes. This visit took place in the clinic where patients got their routine medical care. Patients were offered the opportunity to bring a family member with them. To control attentional effects, ETAU nurses followed up with participants with 9 brief phone calls spaced out over the course of 6 months.

**Fidelity:** Attendance for each TEAM and ETAU contact was recorded. Acceptability for each intervention was assessed at 13 weeks with a brief self-rated survey. Following Fraser et al., [13] fidelity to TEAM was assessed by evaluation of a randomly selected 20% of sessions by non-interventionist study staff to determine if sessions covered relevant constructs and health practices. Each fidelity dimension was rated on a 1–10 scale. Different nursing personnel delivered the TEAM and ETAU interventions to minimize chance of contamination across study arms.

## Measures

Demographic variables and existing medical burden were evaluated at screening/baseline and included medical status, personal and family stroke history. We also collected information on medications for stroke risk factors (type and total number) and queried individuals on difficulty (financial or otherwise) in taking medications or medical care. The Alcohol Use Disorders Identification Test (AUDIT) identified those with problem alcohol use [14–18]. Tobacco use was measured by the Global Adult Tobacco Survey (GATS) questionnaire [19]. Given low rates of tobacco use, smoking status was dichotomized as yes/no.

Participants received modest compensation for completing research assessments.

## Primary outcome

The primary outcome was a change in SBP from baseline to 6 month follow up. All BP assessments used a standardized protocol [20] and measured with an Omron automated sphygmomanometer model HEM 907 with validated accuracy [21].

## Secondary outcomes

a. Blood based biomarkers: Secondary biomarker outcomes included DBP, serum cholesterol and lipids, HbA1c and BMI. Hyperlipidemia was defined as having abnormal/elevated serum cholesterol/lipids. The serum assessments conducted were cholesterol, high-density lipoprotein (HDL), low-density lipoprotein (LDS) and triglycerides. Normal references range for cholesterol were: 0–5.17 mmol/L, for LDL: 0–4.11 mmol/L, for HDL: 1.15–3.3 mmol/L and for triglycerides: 0–2.3 mmol/L. Elevated cholesterol was defined as > 5.2 mmol/L, elevated LDL was defined as > 3.4 mmol/L and elevated triglycerides was defined as > 1.7 mmol/L. For HDL, abnormal values are below the reference range and differ by gender. Abnormal HDL was defined as < 1.0 mmol/L for men and < 1.3 mmol/L for women. Diabetes was defined with an HbA1c level of ≥ 6.5%. Obesity was classified as BMI > 30.

b. Life-style and attitudinal factors: Additional secondary outcomes included dietary sodium measured by a modified dietary questionnaire [14], and physical activity measured by the Global Physical Activity Questionnaire (GPAQ) [15] Self-efficacy was assessed with the General Self-Efficacy measure [14–18]. Medication adherence was measured via the Medication Adherence Report Scale/MARS) [22,23]. We assessed stress using a combined measure of home and workplace stress adapted from the INTERSTROKE study [24]. Depression was assessed with items from the INTERSROKE study.

## Clinician perceptions

As a broad measure of clinician satisfaction, treating (non-study clinicians) were asked to provide their level of agreement/disagreement with the statement: "In my opinion, the program has helped my patient manager their stroke risk(s)"

## Data analysis

Univariate descriptives are provided in Table 1. Primary and secondary outcomes were assessed at baseline, at 13 weeks and 6 months. To test our primary outcome of change from baseline to 6-months we used 2 group by 3-time points repeated measures analysis of variance (RMANOVA) to compare two groups (TEAM vs. ETAU) across the three-time points at baseline, 13 weeks and 6 months SBP (H1), serum cholesterol (H2), and serum HbA1c (H3). Analysis included all who were randomized to TEAM and ETAU and who had data for baseline, 13- and 24-week data. The level of significance was set at $\alpha \leq 0.05$. Given the relatively small amount of missing data, we did not conduct imputation analyses. However, we have compared baseline demographic and clinical characteristics of missing cases and did not find any substantive correlations with missing cases, suggesting no systematic bias in missing data. These findings are consistent with data missing completely at random.

# Results

## Overall sample

A total of 330 individuals were screened with 64 individuals excluded because they did not meet entry criteria or complete screening. There were 250 individuals randomized. Three randomized individuals were later determined to have been erroneously enrolled. Data from these 3 participants were sequestered from the sample, leaving a total of 247 randomized, analyzable participants (122 allocated to TEAM and 125 allocated to ETAU). There were 113 individuals in TEAM and 112 in ETAU at 6-month follow-up (n = 225 in the overall analyzable sample), see Fig 1. The overall attrition rate (removing the 3 sequestered cases) was 8.9% and similar between arms.

**Table 1. Baseline characteristics of the TEAM Uganda sample (N = 247).**

| Variable | Total Sample N = 247 | | TEAM N = 122 | ETAU N = 125 | TEAM N = 122 | ETAU N = 125 | p |
|---|---|---|---|---|---|---|---|
| | N (%) | Mean (SD), range | N (%) | N (%) | Mean (SD) | Mean (SD) | |
| Age (years) | | 55.4 (12.0), 20-90 | | | 55.2 (11.6) | 55.7 (12.5) | .71 |
| Gender | | | | | | | |
| Male | 79 (32.0) | | 38 (31.1) | 41 (32.8) | | | .78 |
| Female | 168 (68.0) | | 84 (68.9) | 84 (67.2) | | | |
| Educational level | | | | | | | |
| None | 18 (7.3) | | 10 (8.2) | 8 (6.4) | | | .13 |
| Primary | 118 (47.8) | | 51 (41.8) | 67 (53.6) | | | |
| Secondary | 87 (35.2) | | 51 (41.8) | 36 (28.8) | | | |
| Tertiary | 24 (9.7) | | 10 (8.2) | 14 (11.2) | | | |
| Marital status | | | | | | | |
| Single | 15 (6.1) | | 7 (5.7) | 8 (6.4) | | | .02 |
| Married | 132 (53.4) | | 76 (62.3) | 56 (44.8)* | | | |
| Separated/Divorced/Widowed | 100 (40.5) | | 39 (32.0) | 61 (48.8)* | | | |
| Currently employed/working[1] | 161 (65.2) | | 82 (67.2) | 79 (63.2) | | | .51 |
| Residency status | | | | | | | |
| Rural | 53 (21.5) | | 25 (20.5) | 28 (22.4) | | | .16 |
| Urban | 111 (44.9) | | 49 (40.2) | 62 (49.6) | | | |
| Suburban | 83 (33.6) | | 48 (39.3) | 35 (28.0) | | | |
| Family stroke history | 50 (20.2) | | 25 (20.5) | 25 (20.0) | | | .92 |
| Stroke risk factors at screenings | | | | | | | |
| Systolic BP ≥ 140 | 247 (100) | | | | | | |
| Diabetes | 72 (29.1) | | 33 (13.4) | 39 (31.2) | | | .47 |
| Hyperlipidemia | 154 (62.3) | | 79 (63.2) | 75 (61.5) | | | .78 |
| Obesity | 98 (39.7) | | 51 (41.8) | 47 (37.6) | | | .50 |
| Current smoker | 3 (1.2) | | 1 (0.8) | 2 (1.6) | | | .58 |
| Sedentary lifestyle | 58 (23.5) | | 25 (20.5) | 33 (26.4) | | | .27 |
| Problem alcohol use | 26 (10.5) | | 12 (9.8) | 14 (11.2) | | | .73 |
| Personal history of stroke or TIA | 81 (32.8) | | 36 (29.5) | 45 (36.0) | | | .28 |
| Total number stroke risk factors at screening | | 3.0 (0.9), 2-6 | | | 2.9 (0.9) | 3.1 (1.0) | .17 |
| Receiving medication for HTN (at least 1 drug) | 242 (98.0) | | 119 (97.5) | 123 (98.4) | | | .63 |
| Stroke risk biomarkers | | | | | | | |
| Systolic BP at screen | | 158.1 (14.9) 140-216 | | | 158.4 (14.8) | 157.8 (15.0) | .74 |
| Diastolic BP at screen | | 95.4 (11.4) 71.5-132 | | | 95.3 (10.6) | 95.5 (12.2) | .92 |
| Systolic BP at baseline | | 143.0 (19.8), 94.5-206 | | | 145.2 | 140.8 | .08 |
| Diastolic BP at baseline | | 89.3 (14.0), 61-136 | | | 91.1 (14.6) | 87.5 (13.2) | .04 |
| Serum cholesterol Proportion with elevated value | 61 (24.7) | 4.5 (1.1), 1.7-7.1 | | | 4.5 (1.2) | 4.5 (1.1) | .86 |
| Serum HDL, proportion with abnormal (low) value | | | | | | | |
| Men (n = 79) | 25 (31.6) | 1.6 (2.8), | | | 1.3 (0.6) | 1.3 (0.6) | .97 |
| Women (n = 166) | 90 (54.2) | .3-4.4 | | | | | |
| Serum LDL, proportion with elevated value | 82 (33.2) | 3.0 (1.3), 4-8.3 | | | 3.0 (1.3) | 3.1 (1.4) | .73 |

*(Continued)*

**Table 1.** (Continued)

| Variable | Total Sample N = 247 | | TEAM N = 122 | ETAU N = 125 | TEAM N = 122 | ETAU N = 125 | p |
|---|---|---|---|---|---|---|---|
| | N (%) | Mean (SD), range | N (%) | N (%) | Mean (SD) | Mean (SD) | |
| Serum triglycerides, proportion with elevated value | 84 (34.0) | 1.6 (1.0), 3-7.7 | | | 1.6 (0.9) | 1.7 (1.1) | .23 |
| HbA1c, proportion with diabetes | | 6.1 (1.9), 2.0-15.0 | | | 6.2 (1.9) | 6.0 (1.9) | .47 |
| BMI, proportion with obesity | 98 (39.7) | 28.7 (5.8), 16.8-45.5 | | | 29.3 (5.9) | 28.2 (5.7) | .14 |
| BMI, sub-group with obesity | | 34.4 (3.9), 30.0-45.5 | | | 34.8 (4.0) | 34.0 (3.7) | .27 |
| Life-status/style and attitudinal factors | | | | | | | |
| Dietary Sodium Questionnaire | | 13.3 (4.8) | | | 12.9 (5.0) | 13.7 (4.6) | 0.18 |
| Global Physical Activity Questionnaire (GPAQ) | | 940.3 (1025.5), 5-5740 | | | 963.6 (1097.3) | 916.6 (951.2) | .73 |
| Current Smoker (N/%) *** | 4 (1.6) | | 2 (1.6) | 2 (1.6) | | | .98 |
| AUDIT total, proportion with problem drinking behavior ** | 8 (4.9) | 1.0 (3.5), 0-30 | | | 1.0(3.7) | 1.0 (3.4) | .87 |
| General Self-Efficacy | | 29.8 (5.6),13- 40 | | | 29.4 (5.5) | 30.2 (5.7) | .30 |
| Medication Adherence Report Scale (MARS) | | 8.1 (1.9), 1-10 | | | 8.2 (1.8) | 7.9 (2.0) | .21 |
| Difficulty in accessing | | | | | | | |
| Medications | 202 (81.8) | | 96 (78.7) | 106 (84.8) | | | .21 |
| Source of difficulty | | | | | | | |
| Financial | 206 (83.4) | | 103 (84.4) | 103 (82.4) | | | .57 |
| Transportation | 5 (2.0) | | 1 (0.8) | 4 (3.2) | | | |
| Other | 5 (2.0) | | 3 (2.5) | 2 (1.6) | | | |
| Medical care difficulty | 173 (70.0) | | 86 (70.5) | 87 (69.6) | | | .88 |
| Source of difficulty | | | | | | | |
| Financial | 121 (49.0) | | 63 (51.6) | 58 (46.4) | | | .71 |
| Transportation | 56 (22.7) | | 26 (21.3) | 30 (24.0) | | | |
| Other | 16 (6.5) | | 9 (7.4) | 7 (5.6) | | | |
| Stress instrument | | 6.4 (1.7),3-11 | | | 6.5 (1.6) | 6.4 (1.9) | .58 |
| Depressive symptoms | | 2.5 (2.7), 0-7 | | | 2.4 (2.7) | 2.6 (2.7) | .50 |

[1]Includes self-employed work at home or farming, *Significant differences in subgroups

SD: Standard deviation, HTN: Hypertension, BP: Blood Pressure, BMI: Body Mass Index. Obesity defined as BMI > 30. AUDIT: Alcohol Use Disorders Identification Test. ** Harmful drinking behaviors based on AUDIT score defined as total score > 8. *** Current smoke based on baseline assessment using the Global Adult Tobacco Survey (GATS)

The HDL values for two extreme outlier cases (due to a presumed genetic variant) were not included in means and correlational analysis for this biomarker.

Table 1 shows demographic, clinical and life-style variables from screening and baseline visits. Mean age was 55.4 (12.0), 20–90 years. The majority were women (n = 168, 68%) and nearly half (n = 118, 47.8%) had a primary-level education. Just over 1 in 5, n = 53 (21.5%), lived in rural settings. There were 81 individuals (32.8%) with previous stroke or TIA and 50 (20.2%) with family stroke history. In addition to hypertension, the most common screening stroke risk factors were hyperlipidemia (n = 154, 62.3%) and obesity (n = 98, 39.7%). A scant minority (n = 4, 1.6%) were smokers. The

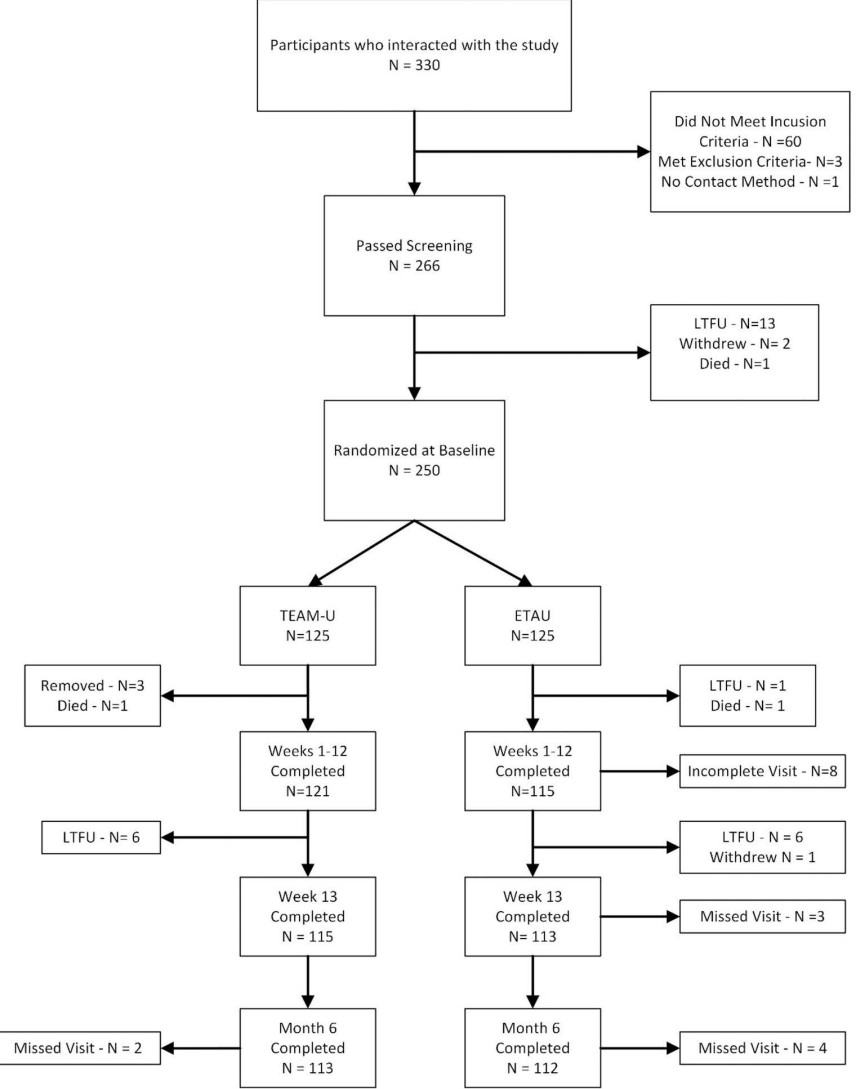

**Fig 1. Consolidated Standards of Reporting Trail (CONSORT) diagram.** Flowchart of participants disposition throughout the study.

overwhelming majority (n = 242, 98%) were prescribed antihypertensive medications. Blood-based biomarkers were within normal group means for serum cholesterol, HDL, LDL and for triglycerides.

## Attendance and safety

Both TEAM and ETAU were relatively well attended. In TEAM there were 106 (86.9%) individuals who attended all 6 group sessions, and the average attendance was 5.6 (SD 1.4) sessions. For the 3 follow-up phone calls, the mean number of phone calls was 2.8 (SD 0.7). In ETAU, for the 9 phone calls with the nurse there were 74 (59.7%) who attended all 9 calls, and the average call attendance was 6.4 (SD 3.6).

Over the course of the trial, there were 5 serious adverse events (SAEs), three deaths and 2 serious medical events (acute kidney failure, sepsis). No adverse events were study related.

## Longitudinal outcomes

Table 2 shows changes in SBP and other biomarkers. In TEAM, SBP went from a baseline value of 145.7 (21.5) to a 6-month follow-up value of 137.4 (18.1) while ETAU respective values were 141.9 (18.4) and 141.1 (21.9), a significant difference favoring TEAM (p = 0.031). Similarly, there was a significant reduction in DBP in TEAM vs. ETAU also favoring TEAM (p = .012). There were no significant differences for TEAM vs. ETAU in total cholesterol, HDL, LDL or triglycerides. Among the sub-set of individuals with elevated serum cholesterol at baseline (n = 28 in TEAM, n = 27 in ETAU) the TEAM group baseline was 5.88 (SD 0.53) and 6-month value was 4.66 (SD 0.68) compared to ETAU baseline of 5.83 (SD 0.47) and 6-month value of 4.86 (SD 0.78), a non-significant difference (p = 0.218.) There was no significant difference on BMI between study arms. For individuals with baseline elevated HA1c (≥6.5) there was no significant difference between TEAM vs. ETAU at 6-months.

**Table 2. Longitudinal outcome of TEAM vs. ETAU baseline to 6-month follow-up on BP and other biomarkers.**

| Variable | Screening | Baseline | 13-weeks | 6-months | Statistic** |
|---|---|---|---|---|---|
| | Mean (SD) | Mean (SD) | Mean (SD) | Mean (SD) | |
| BP | | | | | |
| Systolic (n = 222) | | | | | |
| TEAM (n = 113) | 158.4 (14.8) | 145.7 (21.5) | 137.3 (17.9) | 137.4 (18.1) | **.031** |
| ETAU (n = 109) | 157.8 (15.0) | 141.9 (18.4) | 138.1 (21.1) | 141.1 (21.9) | |
| Diastolic (n = 222) | | | | | |
| TEAM (n = 113) | 95.3 (10.6) | 91.4 (14.6) | 86.5 (12.6) | 86.4 (11.3) | **.012** |
| ETAU(n = 109) | 95.5 (12.2) | 88.3 (13.4) | 86.9 (13.9) | 88.7 (14.0) | |
| **Blood biomarkers** | | | | | |
| Cholesterol (n = 214) | | | | | |
| TEAM (n = 109) | 4.5 (1.2) | NA | 4.3 (1.0) | 4.2 (0.9) | .528 |
| ETAU (n = 105) | 4.5 (1.1) | | 4.3 (1.0) | 4.4 (0.9) | |
| HDL (n = 212) | | | | | |
| TEAM (n = 108) | 1.3 (0.6) | NA | 1.2 (0.5) | 1.2 (0.4) | .912 |
| ETAU (n = 104) | 1.3 (0.6) | | 1.3 (0.5) | 1.2 (0.3) | |
| LDL (n = 214) | | | | | |
| TEAM (n = 109) | 2.9 (1.3) | NA | 2.5 (1.1) | 2.5 (0.9) | .624 |
| ETAU (n = 105) | 3.0 (1.4) | | 2.5 (0.9) | 2.7 (0.9) | |
| Triglycerides (n = 214) | | | | | |
| TEAM (n = 109) | 1.6 (0.9) | NA | 1.4 (0.7) | 1.4 (0.6) | .387 |
| ETAU (n = 105) | 1.7 (1.1) | | 1.4 (0.7) | 1.4 (0.7) | |
| HA1c * (n = 66) | | | | | |
| TEAM (n = 30) | 8.7 (2.2) | NA | 7.3 (1.9) | 7.0 (2.1) | .223 |
| ETAU (n = 36) | 8.3 (1.6) | | 7.0 (1.8) | 7.4 (2.4) | |
| Weight | | | | | |
| BMI (n = 218) | | | | | |
| TEAM (n = 110) | 29.3 (5.7) | NA | 29.0 (5.2) | 28.8 (5.0) | .878 |
| ETAU (n = 108) | 28.0 (5.6) | | 27.8 (5.4) | 27.6 (5.3) | |

Screening BP values were not included in longitudinal analysis. Data is presented here for informational purposes only

* HA1c only presented for those classified as having diabetes (baseline HA1c ≥ 6.5%)

Obesity was classified using body mass index (BMI) with a value of > 30

** Data analysis includes only individuals that had data at all timepoints.

Table 3 shows changes in lifestyle and attitudinal factors. Compared to ETAU, individuals in TEAM had significantly improved physical activity (p = .017), self-efficacy (p < .001) and stress (p = .014). There was no significant difference in dietary salt intake in TEAM vs. ETAU or in depression.

## Participant acceptability/satisfaction

In the TEAM sample, 116 (95%) completed the post-intervention survey. Of these, 100% strongly agreed or agreed that TEAM is useful, covers all/most of the important issues, addresses issues important to their particular situation, and that they would recommend it to others. Most (N = 113, 97.4%) felt that the benefit of TEAM exceeded the burden of participation. With respect to the number of sessions, only 1 individual (0.9%) felt there were too many sessions, while N = 104 (89.7%) felt the number of sessions were just right and N = 11 (9.5%) felt there were too few sessions. With respect to the length of time for each session, N = 7 (6.0%) felt they were too short and N = 109 (94.0%) felt they were just right. No individuals felt the sessions were too long.

In the ETAU sample, 115 (92%) completed the post-intervention survey. Most of the participants strongly agreed or agreed that ETAU was useful (N = 110, 95.7%) and addresses issues important to their situation (N = 95, 82.6%). Just over a quarter (N = 87, 75.7%) felt the benefits of ETAU exceeded the burden of participation and nearly all (N = 113 (98.3%) would recommend it to others.

## Clinician satisfaction

Clinicians were not aware of whether their patients were randomized to TEAM or ETAU. Among 224 clinicians who responded to the survey, the great majority (N = 128, 57.1%) strongly agreed or agreed (N = 82, 36.6%) that the program helped their patient manage stroke risk. Clinician perceptions between TEAM and ETAU were largely similar.

**Table 3. Longitudinal secondary outcomes of baseline to 6-month follow-up in TEAM vs. ETAU among high-risk Ugandans.**

| Variable | Screening | Baseline | 13-weeks | 6-months | Statistic** |
|---|---|---|---|---|---|
| | Mean (SD) | Mean (SD) | Mean (SD) | Mean (SD) | |
| Dietary sodium (n = 247) | | | | | |
| TEAM (n = 122) | 0 (0) | 12.9 (5.0) | 11.6 (6.1) | 11.8 (5.3) | **0.328** |
| ETAU (n = 125) | 0 (0) | 13.7 (4.6) | 10.9 (5.5) | 11.1 (4.6) | |
| GPAQ (n = 208) | | | | | |
| TEAM (n = 109) | 947.1 (1082.0) | | 1210.5 (1097.4) | 1353.9 (1061.4) | **.017** |
| ETAU (n = 109) | 974.5 (929.0) | | 1033.3 (1057.7) | 872.3 (1040.9) | |
| Self-efficacy (n = 222) | | | | | |
| TEAM (n = 113) | | 29.6 (5.5) | 31.8 (4.7) | 32.4 (4.0) | **<.001** |
| ETAU (n = 109) | | 30.4 (5.8) | 29.5 (5.1) | 30.1 (4.8) | |
| MARS (n = 221) | | | | | |
| TEAM (n = 112) | | 8.3 (1.8) | 8.9 (1.5) | 9.2 (1.2) | .672 |
| ETAU (n = 109) | | 7.9 (2.0) | 8.7 (1.6) | 8.8 (1.4) | |
| Stress Instrument (n = 222) | | | | | |
| TEAM (n = 113) | | 6.5 (1.7) | 6.2 (1.7) | 5.8 (1.5) | **.014** |
| ETAU (n = 109) | | 6.4 (1.8) | 6.8 (1.6) | 6.5 (1.5) | |
| Depressive Symptoms * (n = 222) | | | | | |
| TEAM (n = 113) | | 2.4 (2.7) | 1.7 (2.5) | 1.4 (2.2) | .625 |
| ETAU (n = 109) | | 2.7 (2.7) | 1.8 (2.5) | 1.3 (2.3) | |

GPAQ: Global Physical Activity Questionnaire, MARS: Medication Adherence Report Scale

*Depression severity was assessed with the depression items from the INTERSROKE study stress instrument.

** Data analysis includes only individuals that had data at all timepoints.

## Discussion

Stroke rates and stroke-related deaths are increasing rapidly in Africa, which has one of the highest indices of stroke burden in the world [25,26]. This RCT tested a novel risk reduction intervention among high-risk Ugandans. Compared to ETAU, individuals in TEAM had lower blood pressure at 6-month follow-up as well as improvements in physical activity levels, self-efficacy and stress. Strengths of the study include a generalizable sample drawn from rural, suburban and urban settings, a favorable trial attrition rate of < 9%, and a rigorous comparator. The TEAM approach represents a practical and potentially scale-able format which uses nurses already like to be present in primary care settings and individuals with lived experience of stroke risk factors to deliver the intervention.

Two main strategies have been proposed for stroke prevention: a high-risk strategy and a population-based strategy, [27,28] each of which use different approaches to achieve the goal of reducing stroke burden. The TEAM program falls under the category of a high-risk strategy as it identifies and targets populations highly vulnerable to stroke occurrence. The RCT included both those who had never had stroke as well as those with previous stroke (32.8% of the participants). Consistent with the high-risk reduction approach, TEAM focuses on changes in lifestyle as well as promoting appropriate use of medication treatment.

Our original hypothesis, that TEAM would reduce SBP, was confirmed. Observations from the INTERSTROKE study, which involved 5 countries in Africa (Uganda, Mozambique, Nigeria, South Africa and Sudan) found that hypertension was the major modifiable risk factor [24] The Stroke Investigative Research and Educational Network Study (SIREN), a multi-site case–control stroke risk factor study that included 2,118 case–control pairs of African adults in Nigeria and Ghana similarly identified hypertension as a key risk factor [29]. A review by Akinyemi and colleagues noted that hypertension is the prime modifiable driver of the stroke burden in Africa [30].

The top 10 stroke risk factors identified by the INTERSTROKE study included hypertension, dyslipidemia, diabetes, obesity, cardiac causes, smoking, alcohol, poor diet, physical inactivity and psychosocial factors [24]. The TEAM approach addresses all of these. However, in spite of this, we were not able to confirm our hypotheses that TEAM was superior to ETAU in reducing serum cholesterol. It seems likely that this is because group means at baseline fell within normal range for all blood-based biomarkers although we also did not see significant differences between TEAM and ETAU in cholesterol in the small sub-group (22%, N = 55) who had elevated baseline cholesterol.

Our RCT was also unable to confirm the hypothesis that individuals with diabetes randomized to TEAM would have improved glycemic control. Both the TEAM and ETAU groups had substantial improvement in HbA1c. It is possible that the small sample size of people with diabetes (n = 30 for TEAM, n = 36 for ETAU) made it difficult to identify intervention arm differences. Alternatively, since the study did not provide medication treatments for diabetes and a large majority of people noted financial barriers to obtaining medications (over 80%), either intervention's ability to make more substantive impact may have been limited. The Prospective Urban Rural Epidemiological (PURE) study, which recruited 153, 996 adults in countries at various stages of economic development found that secondary prevention is especially problematic in low-income countries and rural areas [31].

With respect to secondary outcomes, compared to ETAU, TEAM appears to have positive effects on physical activity (p = .017), self-efficacy (p < .001) and stress (p = .014). Other studies on stroke in SSA have emphasized the need for culturally sensitive and acceptable community-based educational approaches [7]. The format of the TEAM program, which features people with lived experience may help minimize interpersonal and logistic stroke risk management challenges. Anecdotal input from TEAM participants suggests that including people who had stroke in groups with people who have never had stroke provides context and perspective for all.

Akinyemi and colleagues proposed a "stroke quadrangle" for reducing stroke burden in Africa, with 4 key components or "pillars" that include stroke surveillance, prevention, acute care and rehabilitation [30]. Tsshe TEAM program represents an integrated approach that targets both prevention in those at high stroke risk and recovery for those who have had stroke. Strong acceptability and perceived value for TEAM is suggested by high program attendance and positive clinician and patient feedback.

In spite of our positive RCT findings, generalizability could still be limited. The 3-site enrollment did not represent all regions of the country. Most individuals were receiving medication treatment and findings may not apply to those with untreated risk factors. Additionally, the RCT was conducted during the COVID-19 pandemic which could have impacted engagement.

In conclusion, there remains a need for effective and practical approaches to reduce stroke burden in SSA. Based on RCT outcomes, inclusion of the TEAM approach in primary care settings seems to be a pragmatic and effective way to potentially reduce stroke burden in Uganda.

## Supporting information

**S1 File. Consolidated Standards of Reporting Trail for social and psychological interventions (CONSORT-SPI) checklist for study.**
(DOCX)

**S2 File. TEAM study protocol.**
(PDF)

**S3 File. Reducing stroke burden protocol for a randomised for a randomised controlled trial.**
(DOCX)

## Acknowledgments

The authors would like to thank all the study participants with stroke or those with risk factors who participated in the study.

## Author contributions

**Conceptualization:** Mark Kaddumukasa, Christopher J. Burant, Elly T. Katabira, Martha Sajatovic.

**Data curation:** Mark Kaddumukasa, Martin Kaddumukasa, Scovia Nalugo Mbalinda, Josephine Najjuma, Jane Nakibuuka, Joy Yala, Levicatus Mugenyi, Christopher J. Burant, Shirley Moore, Elly T. Katabira, Martha Sajatovic.

**Formal analysis:** Scovia Nalugo Mbalinda, Josephine Najjuma, Jane Nakibuuka, Joy Yala, Levicatus Mugenyi, Christopher J. Burant, Shirley Moore, Martha Sajatovic.

**Funding acquisition:** Carla Conroy, Elly T. Katabira, Martha Sajatovic.

**Investigation:** Mark Kaddumukasa, Martin Kaddumukasa, Scovia Nalugo Mbalinda, Josephine Najjuma, Jane Nakibuuka, Joy Yala, Levicatus Mugenyi, Shirley Moore.

**Methodology:** Mark Kaddumukasa, Martin Kaddumukasa, Scovia Nalugo Mbalinda, Josephine Najjuma, Jane Nakibuuka, Joy Yala, Levicatus Mugenyi, Shirley Moore, Elly T. Katabira, Martha Sajatovic.

**Project administration:** Mark Kaddumukasa, Josephine Najjuma, Doreen Birungi, Carla Conroy, Joy Yala, Elly T. Katabira, Martha Sajatovic.

**Resources:** Scovia Nalugo Mbalinda, Josephine Najjuma, Doreen Birungi, Carla Conroy, Joy Yala, Levicatus Mugenyi, Christopher J. Burant, Shirley Moore, Elly T. Katabira.

**Software:** Joy Yala, Levicatus Mugenyi, Christopher J. Burant.

**Supervision:** Mark Kaddumukasa, Martin Kaddumukasa, Scovia Nalugo Mbalinda, Josephine Najjuma, Doreen Birungi, Carla Conroy, Levicatus Mugenyi, Christopher J. Burant, Martha Sajatovic.

**Validation:** Mark Kaddumukasa, Martin Kaddumukasa, Scovia Nalugo Mbalinda, Josephine Najjuma, Doreen Birungi, Carla Conroy, Levicatus Mugenyi.

**Writing – original draft:** Mark Kaddumukasa, Martin Kaddumukasa, Scovia Nalugo Mbalinda, Josephine Najjuma, Jane Nakibuuka, Doreen Birungi, Carla Conroy, Joy Yala, Levicatus Mugenyi, Christopher J. Burant, Shirley Moore, Elly T. Katabira, Martha Sajatovic.

**Writing – review & editing:** Mark Kaddumukasa, Josephine Najjuma, Jane Nakibuuka, Levicatus Mugenyi, Christopher J. Burant, Shirley Moore, Elly T. Katabira, Martha Sajatovic.

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
