## [Decision Letter · Decision Letter 0]

25 Jun 2025

Dear Dr. Kaddumukasa,

Thank you for submitting your manuscript to PLOS ONE. After careful consideration, we feel that it has merit but does not fully meet PLOS ONE’s publication criteria as it currently stands. Therefore, we invite you to submit a revised version of the manuscript that addresses the points raised during the review process.

We look forward to receiving your revised manuscript.

Kind regards,

Toshiki Maeda

Academic Editor

PLOS ONE

Journal Requirements:

“Martha Sajatovic and Elly Katabira received the award. “Research reported in this publication was supported by the National Institute Of Neurological Disorders And Stroke of the National Institutes of Health under Award Number R01NS118544. The content is solely the responsibility of the authors and does not necessarily represent the official views of the National Institutes of Health.”

“Dr. Sajatovic has research grants from Neurelis, Intra-Cellular, Merck, Otsuka, Teva and Alkermes, is a consultant to Alkermes, Otsuka, Janssen, Lundbeck, Teva and Neurelis and has received publication royalties from Springer Press, Johns Hopkins University Press, Oxford Press, and UpToDate.  The other authors have nothing to report.”

5. We note that your Data Availability Statement is currently as follows: All relevant data are within the manuscript and its Supporting Information files.

7. Your ethics statement should only appear in the Methods section of your manuscript. If your ethics statement is written in any section besides the Methods, please move it to the Methods section and delete it from any other section. Please ensure that your ethics statement is included in your manuscript, as the ethics statement entered into the online submission form will not be published alongside your manuscript.

8. Please ensure that you refer to Figure 1 in your text as, if accepted, production will need this reference to link the reader to the figure.

Reviewers' comments:

Reviewer's Responses to Questions

**Comments to the Author**

1. Is the manuscript technically sound, and do the data support the conclusions?

Reviewer #1: Yes

Reviewer #2: Yes

2. Has the statistical analysis been performed appropriately and rigorously?

Reviewer #1: Yes

Reviewer #2: Yes

3. Have the authors made all data underlying the findings in their manuscript fully available?

Reviewer #1: Yes

Reviewer #2: Yes

4. Is the manuscript presented in an intelligible fashion and written in standard English?

Reviewer #1: Yes

Reviewer #2: Yes

Reviewer #1: The manuscript is generally well written from a statistical perspective. The protocol was excellent as supplemental information providing the detail needed to explain the statistical design and analysis of this project. This included deriving a valid sample size and embellishment of the analysis detail in the manuscript. The sample size for the primary analysis is certainly adequate and accommodating of the secondary considerations of the trial. The ANOVA and MANOVA approach with repeated measures and added covariates certainly covers the routine aspects of the analysis expectations in this design. The authors did well in identifying the limitations of the analysis and the Team approach. There are several questions to be addressed:

1.What is 3wave? Is that simply the times, Baseline, 13 weeks and six months or something else?

2. The investigators note that data that remain missing despite their retention efforts will be accommodated in their analyses and their impact evaluated through sensitivity analyses. Assuming MARS appears reasonable. However, why was there no consideration of the usual imputation strategies common in this case? Where is the sensitivity analysis, if any?

3. Also, the authors note that they will also consider an AR (1) covariance model and compare model fits There is really no discussion of covariate or confounder impact or model comparisons.

Reviewer #2: The manuscript presents a well-conducted randomized controlled trial aimed at reducing stroke risk factors in high-risk individuals in a developing country setting. The study is technically sound, well written, and provides a pragmatic approach for mitigating stroke risk factors through an effective, community-based intervention.

Recommendation: Minor revisions

Specific Comments:

Line 95: It appears the authors intended to report a reduction in systolic blood pressure (SBP) and total serum cholesterol compared to baseline at 24 weeks, rather than the reverse. Please clarify this to avoid confusion.

Intervention Description: The role and background of the PEDs (presumably stroke/TIA survivors acting as peer educators or supporters) should be more clearly described in the interventions section. This would enhance clarity and help readers better understand the implementation strategy.

**Do you want your identity to be public for this peer review?** For information about this choice, including consent withdrawal, please see our Privacy Policy

Reviewer #1: No

Reviewer #2: No

---

## [Author Response · Author response to Decision Letter 1]

17 Jul 2025

7/9/2025

Dear Toshiki Maeda

Academic Editor

PLOS ONE

Re: Response to editorial and reviewer’s comments for PONE-D-25-28874; A 6-month, prospective randomized controlled trial of the TargetEd MAnageMent (TEAM) Intervention vs. enhanced treatment as usual among Ugandans at risk for stroke.

On behalf of the authors, I would like to thank you for the quick review and comments to improve our paper. We have made the suggested changes and corrections as shown below. We have resubmitted the manuscript in track changes and clean version for consideration.

We have included a rebuttal letter responding to each of the points raised by the academic editor and reviewers. .

Journal Requirements:

We have revised the manuscript to meet the journal requirements including those for file naming.

The PLOS questionnaire on inclusivity in global research has been completed and submitted.

“Martha Sajatovic and Elly Katabira received the award. “Research reported in this publication was supported by the National Institute Of Neurological Disorders And Stroke of the National Institutes of Health under Award Number R01NS118544. The content is solely the responsibility of the authors and does not necessarily represent the official views of the National Institutes of Health.”

The amended role of the funder has been included in the cover letter as indicated below.

“Dr. Sajatovic has research grants from Neurelis, Intra-Cellular, Merck, Otsuka, Teva and Alkermes, is a consultant to Alkermes, Otsuka, Janssen, Lundbeck, Teva and Neurelis and has received publication royalties from Springer Press, Johns Hopkins University Press, Oxford Press, and UpToDate. The other authors have nothing to report.”

The competing interest statement has been corrected as indicated below.

“Dr. Sajatovic has research grants from Neurelis, Intra-Cellular, Merck, Otsuka, Teva and Alkermes, is a consultant to Alkermes, Otsuka, Janssen, Lundbeck, Teva and Neurelis and has received publication royalties from Springer Press, Johns Hopkins University Press, Oxford Press, and UpToDate. The other authors have nothing to report. This does not alter our adherence to PLOS ONE policies on sharing data and materials.” (as detailed online in our guide for authors http://journals.plos.org/plosone/s/competing-interests)”

5. We note that your Data Availability Statement is currently as follows: All relevant data are within the manuscript and its Supporting Information files.

We have included this statement in the manuscript. We will make the data and associated documentation available to users only under a data-sharing agreement (DUA) that provides for: (1) a commitment to using the data only for research purposes and not to identify any individual participant; (2) a commitment to securing the data using appropriate computer technology; and (3) a commitment to destroying or returning the data after analyses are completed. Beyond this, all the information generated from the various study data sets will be made available to the global community in open access journals indexed in pub med or via the internet. Qualified investigators can contact the study lead investigators: Martha.sajatovic@uhhospitals.org or kaddumark@yahoo.co.uk for execution of an appropriate DUA. See page 24, lines 442 – 450.

This statement has been deleted from the text, see page 19, line 330.

7. Your ethics statement should only appear in the Methods section of your manuscript. If your ethics statement is written in any section besides the Methods, please move it to the Methods section and delete it from any other section. Please ensure that your ethics statement is included in your manuscript, as the ethics statement entered into the online submission form will not be published alongside your manuscript.

The ethics section has been moved to the methods section, see page 8, lines 138 – 144.

8. Please ensure that you refer to Figure 1 in your text as, if accepted, production will need this reference to link the reader to the figure.

Figure 1 is included in the text, see page 12, line 234 – 235. .

The captions for the supporting files have been included at the end of the manuscript. See page 23, lines 417 – 419.

We have reviewed our reference list and it’s complete and correct.

Reviewers' comments:

Reviewer's Responses to Questions

Comments to the Author

6+2

1. Is the manuscript technically sound, and do the data support the conclusions?

Reviewer #1: Yes

Reviewer #2: Yes

2. Has the statistical analysis been performed appropriately and rigorously?

Reviewer #1: Yes

Reviewer #2: Yes

3. Have the authors made all data underlying the findings in their manuscript fully available?

Reviewer #1: Yes

Reviewer #2: Yes

4. Is the manuscript presented in an intelligible fashion and written in standard English?

Reviewer #1: Yes

Reviewer #2: Yes

5. Review Comments to the Author

Reviewer #1: The manuscript is generally well written from a statistical perspective. The protocol was excellent as supplemental information providing the detail needed to explain the statistical design and analysis of this project. This included deriving a valid sample size and embellishment of the analysis detail in the manuscript. The sample size for the primary analysis is certainly adequate and accommodating of the secondary considerations of the trial. The ANOVA and MANOVA approach with repeated measures and added covariates certainly covers the routine aspects of the analysis expectations in this design. The authors did well in identifying the limitations of the analysis and the Team approach. There are several questions to be addressed:

1.What is 3wave? Is that simply the times, Baseline, 13 weeks and six months or something else?

Response: 3 waves refers to the 3 time points of Baseline, 13 weeks, and six months. We have corrected this within the manuscript to time points within the manuscript, see page 12, lines 226 – 229.

2. The investigators note that data that remain missing despite their retention efforts will be accommodated in their analyses and their impact evaluated through sensitivity analyses. Assuming MARS appears reasonable. However, why was there no consideration of the usual imputation strategies common in this case? Where is the sensitivity analysis, if any?

We apologize for any misunderstanding. Given the relatively small amount of missing data, we did not conduct imputation analyses. However, we have compared baseline demographic and clinical characteristics of missing cases and did not find any substantive correlations with missing cases, suggesting no systematic bias in missing data. These findings are consistent with data missing completely at random. We have included this under the data analysis section, see page 12, lines 231 – 235.

3. Also, the authors note that they will also consider an AR (1) covariance model and compare model fits. There is really no discussion of covariate or confounder impact or model comparisons.

We apologize for any misunderstanding. Regarding the AR (1) covariance model, we chose to use RMANOVAs as there were a more parsimonious approach to analyzing the trend in the means of the primary outcomes, the major focus of the paper.

Reviewer #2: The manuscript presents a well-conducted randomized controlled trial aimed at reducing stroke risk factors in high-risk individuals in a developing country setting. The study is technically sound, well written, and provides a pragmatic approach for mitigating stroke risk factors through an effective, community-based intervention.

Recommendation: Minor revisions

Specific Comments:

Line 95: It appears the authors intended to report a reduction in systolic blood pressure (SBP) and total serum cholesterol compared to baseline at 24 weeks, rather than the reverse. Please clarify this to avoid confusion.

This has been corrected to reflect this, see lines 92 – 95.

“A 6-month, uncontrolled, prospective pilot study to establish feasibility, acceptability and preliminary efficacy of TEAM in Ugandans at high risk for stroke showed TEAM participants had significant reductions from baseline in mean systolic blood pressure (SBP) and in total serum cholesterol to 24-weeks levels.(9)”

Intervention Description: The role and background of the PEDs (presumably stroke/TIA survivors acting as peer educators or supporters) should be more clearly described in the interventions section. This would enhance clarity and help readers better understand the implementation strategy.

This section has been rewritten, see page 9, lines 153 -157.

6. PLOS authors have the op

---

## [Decision Letter · Decision Letter 1]

5 Aug 2025

A 6-month, prospective randomized controlled trial of the TargetEd MAnageMent (TEAM) Intervention vs. enhanced treatment as usual among Ugandans at risk for stroke

PONE-D-25-28874R1

Dear Dr. Kaddumukasa,

We’re pleased to inform you that your manuscript has been judged scientifically suitable for publication and will be formally accepted for publication once it meets all outstanding technical requirements.

Kind regards,

Toshiki Maeda

Academic Editor

PLOS ONE

Additional Editor Comments (optional):

Reviewers' comments:

Reviewer's Responses to Questions

**Comments to the Author**

Reviewer #1: All comments have been addressed

2. Is the manuscript technically sound, and do the data support the conclusions?

Reviewer #1: (No Response)

3. Has the statistical analysis been performed appropriately and rigorously?

Reviewer #1: (No Response)

4. Have the authors made all data underlying the findings in their manuscript fully available?

Reviewer #1: (No Response)

5. Is the manuscript presented in an intelligible fashion and written in standard English?

Reviewer #1: (No Response)

Reviewer #1: (No Response)

**Do you want your identity to be public for this peer review?** For information about this choice, including consent withdrawal, please see our Privacy Policy

Reviewer #1: No

---

## [Editor Report · Acceptance letter]

PONE-D-25-28874R1

PLOS ONE

Dear Dr. Kaddumukasa,

I'm pleased to inform you that your manuscript has been deemed suitable for publication in PLOS ONE. Congratulations! Your manuscript is now being handed over to our production team.

Kind regards,

on behalf of

Dr. Toshiki Maeda

Academic Editor

PLOS ONE